# Influence of Social Safety Capital on Safety Citizenship Behavior: The Mediation of Autonomous Safety Motivation

**DOI:** 10.3390/ijerph17030866

**Published:** 2020-01-30

**Authors:** Junjie Zhang, Huaiyuan Zhai, Xiangcheng Meng, Wanxue Wang, Lei Zhou

**Affiliations:** 1School of Economics and Management, Beijing Jiao Tong University, Beijing 100044, China; 18120646@bjtu.edu.cn (J.Z.); 18125665@bjtu.edu.cn (W.W.); 19125682@bjtu.edu.cn (L.Z.); 2School of System Engineering and Engineering Management, City University of Hong Kong, Hong Kong, China; xcmeng3-c@my.cityu.edu.hk

**Keywords:** safety citizenship behavior, social capital, motivation theory

## Abstract

In recent years, the safety issue of construction workers has become a research hotspot, and many researchers have achieved results in the impact of safety behavior regarding China’s construction industry. However, the existing research about the driving factors of safety citizenship behavior is insufficient. To fill this gap, this paper explores the driving factor of safety citizenship behavior from the perspective of social capital theory. A cross-sectional questionnaire survey, involving 311 Chinese construction workers, was conducted to verify the influence of Social Safety Capital on Safety Citizenship Behavior. The results showed that safety citizenship behavior made by workers was significantly related to social safety capital. Autonomous safety motivation mediated the relationships between social safety capital and safety citizenship behavior. Further, this research supports the differences between social safety capital and autonomous safety motivation. Specifically, the paper found that social safety capital had the largest regression coefficient for participation of suggestion-making, and autonomous safety motivation had the largest regression coefficient for the relationship between superior and subordinate by multiple regression analysis.

## 1. Introduction

With the implementation of China’s 13th Five-Year Plan, Chinese construction enterprises have obtained unprecedented achievements. Behind these achievements, numerous accidents caused by the unsafe behavior of staff have brought economic and spiritual losses to China and the workers’ families. According to the Ministry of Housing and Urban–Rural Development of China, in 2018 [1], compared with the data of 2017, the sum of accidents increased by 42 cases (6.1%), while the number of deaths increased by 33 (4.1%). Among them, there were 22 major and above accidents, which killed 87 people; these accidents in 2018 decreased by 1 (4.3%), while the number of deaths decreased by 3 (3.3%). From the 2018 data, the safety situation of construction in China is still poor, in which the number of accidents and the number of deaths are still too high. Accidents still require efforts to reduce their occurrence. Therefore, reducing such accidents is still of great significance for construction projects. To solve the above-mentioned problem, Neal et al. [2] have proposed a safe participation from the perspective of the subjectivity of the actors. Hofmann et al. [3] have emphasized the human subjective initiative by proposing safety citizenship behavior (SCB), which also emphasizes the subjective initiative of the staff and safety participation, which are unaffected by the remuneration system and job responsibilities. Burt et al. [4] have proven that the improvement of SCB leads to the improvement of project safety performance under the premise of risk-free preference of project management and construction parties. The academic community has set off a research boom on SCB. Incentives for SCB affect the factors of quantity and quality of SCB. These questions have become hot issues in the industry, which require urgent solutions.

However, many researchers focus their research on the external organizational environment of employees. Temminck et al. [5] have found that organizational commitments can regulate the relationship between organizational support and organizational citizenship behavior directed towards the environment. Lyu et al. [6] and Griffin et al. [7] have agreed the impact of safety climate and SCB. In related research on leadership, Newman et al. [8] have provided that the leadership–member exchange relationship (LMX) plays an important role in organizational citizenship behavior. Matteo et al. [9] have shown that when leaders and subordinates are in agreement about the quality of their LMX relationship, organizational citizenship behavior is maximized. Townsend et al. [10] have proven that LMX is negatively correlated with retaliation behavior, which has an adverse effect on the company’s safety performance. For research from the perspective of employees, only Matteo et al. [11,12] thought of the mediating role of emotional commitment and psychological ownership in organizational support and SCB. Both areas of research are relatively developed in isolated ways and have no direct or strong links. Therefore, this paper combines these two areas to study SCB, focusing on the process of internal psychology brought by the external organizational environment. The social capital theory is combined in the research model, and has been widely discussed regarding its importance in interpersonal communication (such as safety communication, safety goal, and safety trust) for behavior [13]. 

## 2. Materials and Methods

### 2.1. Materials

In psychology, the psychological driver of human behavior is called motivation. Huang et al. [14] found that the willingness to buy strongly predicts the buying behavior. Motivation, as a special willingness, can be reasonably considered, and may also affect behavior. The reasons for choosing social motivation and personal motivation as research factors will be explained below.

#### 2.1.1. Social Capital

This paper puts the behavior subject into a special work-related working relationship network, which comprises all personnel (including leaders and workers) who are in contact with staff, considering the mechanism of the impact of social motivation derived from the work relationship network on SCB.

In the study of behaviors of relational networks, social capital theory has received extensive attention. Nahapiet et al. [15] defined social capital as “the sum of available real and potential resources embedded in a network of relationships owned by individuals,” stating that social capital has three dimensions, namely, structure, relationship, and cognition. After Coleman et al. [16] found that social capital can promote the improvement of individual behavior, the positive effect on collective behavior was also proved [15,17,18,19]. Chiu et al. [20] have further applied this theory to explain knowledge-sharing behavior, which involves the group-based and active participation behaviors affected by the relationship network. Chow et al. [21] have proven that the relevant factors of the three dimensions of social capital affect knowledge-sharing behavior. Li et al. [22] have found that trust from colleagues plays a role in regulation. According to Neal’s definition, SCB is group-based behavior, which is affected by organizational support, leadership–member exchange relationships, and safety climate, Therefore, it is proposed that SCB is similar to knowledge-sharing behavior. Since social capital is aimed at people in a social network, which includes working relationships, blood relationships, and relationships among friends, relatives, and coworkers [23], this paper reasonably proposes a hypothesis from the perspective of the relationship between existing social capital and knowledge-sharing behavior. 

**Hypothesis** **(H1).**
*Social safety capital can predict safety citizenship behavior.*


Table 1 illustrates the important literature on the three dimensions of social capital. The results show that social networks, social trust, and common goals are often used to measure the performance of structural, relational, and cognitive dimensions. Based on social safety capital, the “common goal” was renamed “safety goal” for research issues; “social trust” was renamed “safety trust”; and “social network” was renamed “safe communication.”

#### 2.1.2. Autonomous Safety Motivation 

Safety motivation has become one of the research subjects of this paper, as an indicator of the motivation of workers’ safety behavior. Neal et al. [28] have defined safety motivation as “the willingness of the employees to perform their work in a safe manner and the individual motivation to demonstrate safety behavior.” Ryan et al. [29] have divided safety motivations into autonomous safety motivations and controlled safety motivations. Autonomous safety motivations involve individuals who engage in certain behaviors voluntarily or according to their own interests and beliefs. Controlled safety motivations encompass individuals engaged in certain behaviors, due to internal or external pressure. SCB itself is a spontaneous behavior that is unaffected by external rewards, as stated by Hofmann et al. [3]; thus, this paper only selects the autonomic safety motivation of safety motivations. Gagne et al. [30] have highlighted that motivation is an important part of an individual’s dynamical system, the motive force of behavior, and the direct driving force of behavior. In the study of social cognition theory and rational behavior theory, Sultan et al. [31] have provided evidence that the stronger the willingness of people to engage in certain behaviors, the more likely they are to actually perform such behaviors. Sustainable behavior is driven by intrinsic and extrinsic motivations, as stated by Bopp et al. [32]. The results of Zhu et al.’s [33] research show that, although different types of motivation have different effects on different types of creativity, intrinsic motivation and extrinsic motivation do have positive effects on creativity. The data are supported in the study, and motivation affects the specific behavior of the person. The following hypothesis is made.

**Hypothesis** **(H2).**
*Autonomous safety motivation can predict safety citizenship behavior.*


#### 2.1.3. Social Capital and Autonomous Safety Motivation

In addition to individual motivation, social motivation is also an important driver of behavior, regarding whether or not a person performs a certain behavior. Behavior is not only affected by personal motivation, but also by others in the network of one’s relationships. The influence of social motivation on behavior cannot be ignored. Bandura [34] also believed that individual behavior is a product of the influence of its network of relationships. In support of the impact of social motivation on personal behavior, Zohar et al. [35] have proven that personal motivation affects safe behavior. Hsin et al. [36] have shown the important role of social capital and individual motivation for knowledge-sharing behavior. At the same time, people with higher social capital pay more attention to their own safety and the safety of others for preventing or reducing the harm to those who are closely related to them. The following assumptions are made. All hypotheses are shown in Figure 1.

**Hypothesis** **(H3).**
*Social safety capital can predict autonomous safety motivation.*


**Hypothesis** **(H4).**
*Autonomous safety motivation mediates the relationship between social safety capital and safety citizenship behavior.*


### 2.2. Methods

#### 2.2.1. Questionnaire Survey

All of the survey items measuring the latent variables were adopted from previous literatures, with established reliability and validity, as is shown in Table 2. Four dimensions with twelve items were used to measure safety citizenship behavior by Meng et al. [37]. The six items used to measure autonomous safety motivation were all adopted from Ryan et al. [29]. As for the three dimensions of social safety capital, we adapted ten items from Sauk et al. [26], Nicola et al. [27], and Chao et al. [20] to measure social capital. Additionally, all of the measurements used a 5-point Likert scale. Eight latent constructs were represented with associated abbreviations during the statistical analysis to adapt to the software environment (shown in Table 2). The details for the scales are shown in Table A1.

A questionnaire survey was conducted by taking the construction industry as the research context. It is considered that the construction industry is one of the most risky sectors because of the high rates of accidents and injuries [38,39,40,41], while the frontline workers are directly exposed to danger and accidents in the workplace [42]. Therefore, the safety research of construction personnel in the construction industry is particularly important. To obtain data from construction personnel, this paper used an easy-to-spread, space-independent online platform to distribute and collect the questionnaire data as a basis for subsequent data analysis in the description section of the questionnaire by Wright et al. [43]. The purpose and precautions of the questionnaire were explained to the construction personnel, followed by the three main parts of the questionnaire, which contained items for measuring SCB, safety motivation, and social safety capital. As of 18 October 2019, the survey received 311 valid questionnaire results and collected demographic data for each participant, which included gender, educational background, age, and working experience. Table A2 summarizes respondents’ demographic characteristics. Respondents were mostly male (79.4%), which is in line with the fact that the workforce in the Chinese construction industry is male-dominated [44]. Among the respondents, those belonging to the age groups of 20–30 and 31–40 accounted for the largest proportion (53.7% and 23.5%, respectively). A total of 24.8% of the respondents had less than two years of work experience in construction, and 30.5% had more than ten years of experience, which means that they likely had a good understanding about workplace safety issues. The majority of respondents (45%) had completed a bachelor’s degree as their highest education. 

#### 2.2.2. Data Analysis Procedures

Data analysis was conducted after data collection and collation were completed. SPSS 24.0 (IBM, Armonk, New York, USA) and AMOS 24.0 (IBM, Armonk, New York, USA) were used for data processing and statistical analysis. For data processing, it was mainly divided into two parts, as shown in Figure 2.

First, the reliability and validity tests of the scales were conducted to ensure their effectiveness.

The validity of the scale was tested by Cronbach’s alpha.The structural validity was tested by the normalized factor load (β), composite reliability (CR), and average variance extraction (AVE) of the confirmatory factor analysis (CFA) output.The discriminant validity was tested using the fitting indicators of multiple CFA models, including degrees of freedom (χ2/df), Tucker–Lewis Index (TLI), comparative fit index (CFI), standardized root mean squared residual (SRMR), and root mean square error of approximation (RMSEA).Exploratory factor analysis (EFA) was conducted first to extract and synthesize the overlapping parts of the original variables into factors so that it can be confirmed that there is no common method deviation.

Second, the hypotheses were tested using the structural equation modeling technique. 

The hypotheses were then tested using Pearson correlation analysis and the structural equation modeling techniqueTested the mediation effect of the autonomous safety motivation as a mediation variable.Tested the regression model with independent safety motivation and social safety capital as independent variables and the SCB four dimensions as dependent variables.Discussed the differences in different demographic groups.

## 3. Results

### 3.1. Reliability and Validity

The data analysis used the approach suggested by Anderson et al. [45]. The first step was to test the reliability of each scale of the questionnaire, and the second step was to test the validity of the latent variable. The two-step approach assessed the reliability and validity of the questionnaire and then tested the model.

The reliability test of the scale is usually evaluated by Cronbach’s alpha [46,47]. Here, it was greater than 0.70, which refers to an acceptable reliability [48], indicating good internal consistency [49]. The equation of Cronbach’s alpha can be found in reference [50]. As shown in Table 3, the smallest of Cronbach’s alpha is 0.745, which is greater than 0.70. The result indicates that the reliability analysis passed. Thus, the questionnaire can be considered reliable.

The convergence validity of the scale was verified by three criteria: (1) The normalized factor load of all items should be significant and exceed 0.50, while the normalized factor load (β) should ideally be greater than 0.70, according to reference [51]. β is the normalized regression coefficient for using Maximum Likelihood Estimation (MLE). The equation of β can be found in reference [50]. (2) The composite reliability (CR) of each dimension should be greater than 0.70 according to reference [52]. (3) The average variance extracted (AVE) for each dimension should bigger than 0.50 according to references [53]. For the current CFA model, the normalized factor load of all items was higher than 0.50 [53], while the normalized factor load of more than 60% of the items was greater than 0.70, which corresponded to the ideal state. Most of the CR and AVE fulfilled the conditions, along with the individual data quality and the number of questionnaire samples, which were slightly lower than the standard value. However, the difference was small, which could be considered to have convergence. The results are provided in Table 4.

Below is an equation of the indicators:(1)$$\mathrm{CR}=\frac{{\left(\Sigma \beta_{i}\right)}^{2}}{{\left(\Sigma \beta_{i}\right)}^{2}+n-\sum \beta_{i}\beta_{j}\;},\;i\;=1,2,3,\;\ldots ,\;n.\;j=1,2,3,\;\ldots ,\;n.$$
(2)$$\mathrm{AVE}=\frac{\sum \beta_{i}\beta_{j}}{n}\;,\;i\;=1,2,3,\;\ldots ,\;n.\;j=1,2,3,\;\ldots ,\;n.$$

The multi-model CFA method was used to evaluate the discriminant validity of the scale [53,54]: (1) By using the chi-square difference test between each model, which comprised various factors and the benchmark model, the significant discriminant validity was obtained. (2) Determining the fitting index, including degrees of freedom (χ2/df), Tucker–Lewis Index (TLI), comparative fit index (CFI), standardized root mean squared residual (SRMR), and root mean square error of approximation (RMSEA), showed that the fitting index of other models was not remarkable, except for the discriminant validity by Hair et al. [51]. The equation of fitting indicators can be found in reference [50]. Table 3 lists the results of the model comparison test. The comparison with the baseline model was significant, whereas the fitting effect of other models was worse than the baseline model. The test for discriminant validity was acceptable. Therefore, the scale obtained good discriminant validity. 

The eight variables were all placed into an exploratory factor analysis to test the results of the non-rotational factor analysis. A total of 23 factors with a trait root greater than 1 was extracted from the factor analysis. The variance of the first common factor was 32%. No case existed wherein only one factor was precipitated or wherein the interpretation rate of a factor exceeded 40% [55,56]. Therefore, no common method bias exists in this survey, thus the common method deviation test passed.

### 3.2. Correlation Analysis

This study used a Pearson correlation analysis to verify the relationship of the four dimensions of SCB, safety motivation, and the three dimensions of social safety capital. The results are shown in Table 5, in which SCB was significantly and positively correlated with social safety capital and autonomous safety motivation, while autonomous safety motivation was significantly and positively correlated with social safety capital. It can be determined that the eight factors are indeed related, and the specific relationship and the corresponding coefficients are obtained through path analysis and Bootstrap method.

Below is an equation of Mean and Standard Deviation:(3)$$\mathrm{Mean}\;=\;\frac{\sum X_{i}}{n},\;i\;=1,2,3,\;\ldots ,\;n.$$
(4)$$\mathrm{Standard}\;\mathrm{Deviation}\;=\;\frac{\sum {\left(X_{i}-M\right)}^{2}}{n},\;i\;=1,2,3,\;\ldots ,\;n.\;M\;\mathrm{is}\;\mathrm{Mean}.$$

### 3.3. Path Analysis and Mediation Effect

According to the hypothesis model, the structural equation model (SEM) was drawn using AMOS. Given that the SCB and social safety capital were represented by four latent variables, this paper packaged the project for the next step. Compared with the standard, fit indice shows an acceptable fit between the SEM model and the data, in which the next test could be performed. Five goodness-of-fit indices were adopted to evaluate the fitness of measurement and structural models, namely, the chi-square divided by degrees of freedom (χ2/df), Tucker–Lewis Index (TLI), comparative fit index (CFI), standardized root-mean-squared residual (SRMR), and root-mean-square error of approximation (RMSEA), as suggested by Hu et al. [57]. A χ2/df less than 3 suggests a good fit of the model by Markus et al. [58]. The suggested criteria for TLI and CFI were higher than 0.9 by Fang et al. [59]. The RMSEA and SRMR values of less than 0.08 indicate a reasonable fitness of the model by Hooper et al. [60]. The results of SEM show an excellent fit of the measurement model to the data (χ2/df = 2.574, TLI = 0.921, CFI = 0.937, RMSEA = 0.071, and SRMR = 0.054).

As shown in Figure 2, the estimated normalization coefficient indicates that between social safety capital (F1) and autonomous safety motivation (F2) (β = 0.471, *p* < 0.001), between autonomous safety motivation (F2) and SCB (F3) (β = 0.416, *p* < 0.001), and between social safety capital (F1) and SCB (F3), (β = 0.368, *p* < 0.001), both are positive and significant. Therefore, assumptions 1, 2, and 3 are supported. Moreover, as shown in Figure 3, the estimation of the normalization coefficient indicates that the direct impact of autonomous safety motivation on SCB was greater than the direct impact of social safety capital on SCB.

In Hypothesis 4, the mediating effect of autonomous safety motivation on social safety motivation and SCB was discussed. This study follows the four steps of MacKinnon et al. [61], established of mediating effects, which require (1) a significant correlation between social safety capital and autonomous safety motivation and (2) a significant correlation between autonomous safety motivation and SCB. (3) While controlling for the autonomous safety motivation, the social safety capital and SCB are still significantly correlated. (4) Meanwhile, a significant effect exists on the intermediary path between social safety capital and SCB. The mediating effect on social safety capital and SCB was tested by Bootstrap, which does not include 0 in the 95% interval. Thus, the mediation effect was significant by Preacher et al. [62,63]. As shown in Table 6, the confidence interval did not contain 0, and the condition was satisfied. The mediation effect was significant. Thus, Hypothesis 3 was supported. In this study, social safety capital had a direct effect on SCB, as it had a partial mediating effect.

Below is an equation of Mean and Standard Deviation:(5)$$\mathrm{Boot}\;\mathrm{SE}\;=\;\frac{SD_{i}}{\surd n}\;,i=1,2,3,\;\ldots ,\;n.\;\mathrm{SD}\;\mathrm{is}\;\mathrm{Standard}\;\mathrm{Deviation}$$
(6)$$Z\;=\;\frac{\mathrm{Effect}\;\mathrm{value}}{\mathrm{Boot}\;\mathrm{SE}}$$

### 3.4. Difference between Social Safety Capital and Autonomous Safety Motivation

This paper further examined the impact of social safety capital and autonomous safety motivation on SCB by analyzing the data with the four dimensions of SCB as the dependent variable. The results are shown in Figure 4. The coefficient of the relationship between autonomous security motivation and relationship between superior and subordinate (REL) was 0.603 which is the biggest coefficient. The coefficient of the relationship between social safety capital and participation of suggestion-making (SUG) was 0.564 which is the second biggest coefficient.

### 3.5. Demographic Differences

Based on the collected demographic information, different groups were classified in terms of personal data, as shown in Table 7. Analysis of Variance (ANOVA) was then used to analyze subgroup differences to identify the particular samples that need to be given extra attention.

By analyzing the significant differences between the four dimensions of SCB in different populations, it was found that the education and age of workers is an important factor that both have significant differences in the three dimensions of SCBs. In the analysis of variance with mutual aid among the workers (HEL) as the dependent variable, significant differences were found in gender, age, education, and working hours. In a past study, Peng et al. [64] found a positive relationship between safety knowledge and attitude towards safety behavior when conducting safety behavior of older construction workers. The age of workers is the important factor for SCBs. This has attracted the attention of scholars. Aryal et al. [65] took workers aged 15–24 as the research subject to conduct occupational health research, and Peng et al. [64] used construction workers aged 50 and over to investigate safety behavior.

This paper divides all the data into the following four parts according to the statistical data of working years and tests the hypothetical model, respectively. The results are shown in Table 8. As for the results of the regression coefficient and significance level test, only the results of the first group (work experience < 2) were significantly different from the results of the full sample model. It was found that working hours are strongly related to age, that is, employees with fewer working years are likely to be younger (β = 0.672, Sig. = 0.000). The occurrence of severe/fatal accidents among younger workers is half as much as that of older ones [66]. The young workers have fully realized the necessity and seriousness of safe work through accident reports on the Internet and courses in schools. There is no significant correlation between social safety capital and safety citizenship behavior.

## 4. Discussion

The results show that autonomous safety motivation has a partial intermediary effect on the impact of social safety capital and SCB. Social safety capital can directly predict SCB, while social safety capital can also predict SCB indirectly through autonomous safety motivation. In summary, the changes in social safety capital, while affecting SCB, can also affect the autonomous safety motivation by using the autonomous safety motivation as a channel of influence, which subsequently affects SCB.

### 4.1. Mediation Effect

This study found that social safety capital is positively correlated with SCB, while autonomous safety motivation has a mediating effect between social safety capital and SCB. In addition, the results support social cognitive theory, which states that human behavior is jointly influenced and controlled by social networks and human cognition, as stated by Bandura et al. [34]. Social safety capital, one of the research subjects of this paper, refers to the sum of all the resources that people can use in the working relationship network. It is a derivative of a kind of social network, which reflects the social network situation to a certain extent. Autonomous safety motivation as a special behavioral intention, to a certain extent, also shows people’s partial understanding of security.

In the first phase of the mediating effect (i.e., social safety capital → autonomous safety motivation), the results show that social safety capital can positively influence the autonomous safety motivation of workers. For instance, users can obtain a positive impact between the three dimensions of social capital and intentions while continuously using Facebook fan pages [67]. Social safety capital comes from social capital and is one of the indicators of social networks. It reflects the interpersonal relationship with internal and external personnel [68]. Social safety capital scores indicate that respondents believe that they maintain advantage in connecting with members of the working relationship network in both high quality and high quantity. The higher the score, the more the employees give support, and the higher level of support that they have. Meanwhile, the respondents are supported by numerous people. Thus, their social support network is intensive. In the study of criminal fears, intensive social support networks actually increase an individual’s anxiety about individual safety, rather than reducing it. People who receive higher social support from others may have increased worry that accidents in their surroundings may endanger their own safety and that of their relatives and friends, compared with those who lack social support [69]. In the construction of buildings, staff will have a stronger psychological demand for safe work due to high social safety capital, thus positively affecting autonomous safety motivation.

For the second phase of the mediation model (i.e., autonomous safety motivation → SCB), this study shows that autonomous safety motivation can positively predict the SCB of workers. In previous studies, the definition of safety motivation can be used to understand autonomous safety motivation. A causal relationship exists between behaviors, as outlined by Neal et al. [2]. Chen et al. [70] further proved that the stronger the employee’s safety motivation, the more employees are willing to conduct safety behavior after ensuring safety compliance and safety participation. A positive relationship between safety motivation and safety behavior was obtained by Vinodkumar et al. [71]. At the same time, the results of this study are consistent with social cognitive theory and self-determination theory.

### 4.2. Difference between Social Safety Capital and Autonomous Safety Motivation

In this study, autonomous safety motivation and social safety capital were used as independent variables, whereas the four dimensions of SCB were used as the dependent variables to construct the SEM. The following results were found: In the model with autonomous safety motivation as the independent variable, the independent variable had the greatest influence on the relationship between superiors and subordinates (REL), after which the Mutual aid among the workers (HEL) and self-control (SEL) were observed to have insignificant differences. The least affected was the participation of suggestion-making (SUG). When social safety capital was the independent variable, the independent variables had the greatest influence on the participation of suggestion-making (SUG); mutual aid among the workers (HEL) and self-control (SEL) were second. The least affected was the relationship between superiors and subordinates (REL). Compared with the model with autonomous safety motivation as the independent variable, the order of influence was the opposite. 

The impact of autonomous safety motivation is reflected by examining the individual’s safety work necessity and self-blame recognition of unsafe work. The higher scores in the autonomous safety motivation items represent the staff’s belief that safety work is crucial. Based on Maslow’s theory of demand, workers are eager for their own efforts to be recognized by others [72]. Similarly, workers who value work safety make their own efforts and hope to be recognized by others. In the working relationship network, this kind of recognition comes from its own superiors. Therefore, when autonomous safety motivation reaches a certain level, particular SCB factors will be provoked and enhanced, such as the relationship between superiors and subordinates (REL), which is closely related to leadership.

The participation of suggestion-making (SUG) differs from the others. It is a factor with a wide range of benefits, by which a large number of people are affected. Social safety capital, as a social motivation from the social network of workers, reflects the sum of resources available to someone in the work relationship network. Studies have shown that extroverts can have more social relations and available resources than introverts, highlighted by Guagnano et al. [73]. Extroverts are considered likely to have either a direct or indirect contact with multiple people in the working relationship network due to their communication skills, thus building an intensive social network. Therefore, people with higher social safety capital are likely to have a sense of responsibility, which encourages them to consider the safety of staff members and consciously contribute ideas and suggestions in the team, resulting in higher scores in suggestions.

## 5. Conclusions

This study aimed to use an established scale to analyze the relationship between social safety capital and autonomous safety motivation. Relying on an online questionnaire survey, this paper obtained data from 311 Chinese construction workers from all over China. The reliability and effectiveness of this survey were tested using CFA-related data, in which satisfactory results were obtained. The results also prove that no evident common method bias exists. Using AMOS to construct the SEM and analyze the data, (1) According to the theory of social capital and social cognition, an intermediary model was established between social safety capital and safety citizenship behavior with autonomous safety motivation as the mediation variable, which passed the test of Bootstrap method. (2) In the regression model with autonomous safety motivation and social safety capital as independent variables, and with the four dimensions of SCB as dependent variables, it was found that the order of the impact of the two different motivations is opposite. 

### 5.1. Contribuctions

The contribution of this article is twofold. First, social safety capital is an important factor in promoting SCB. In previous studies on SCB, some researchers have been more concerned with the impact of leadership–member exchange, leadership behavior, and organizational behavior, whereas other researchers have emphasized individual emotional needs, such as organizational support and emotional commitment. Both areas of research are independently developed and have no potential links. This study combined these two areas to study SCB, focusing on the process of internal psychological changes brought by external changes. The comprehensive model shows social safety capital in interpersonal communication (such as safety communication, safety goal, safety trust), which enhances individual autonomous safety motivation, thus increasing their tendency to SCB. Second, based on the concept of social capital in the past, this paper comprehensively considered the working environment and characteristics of construction workers, and creatively reduced the interpersonal network in social capital to the working relationship network and social capital to the social safety capital in the working relationship network.

### 5.2. Theoretical Implications and Practical Implications

This research has several theoretical contributions for both behavioral theories and safety research. First, this study took the safety citizenship behavior of some construction workers in China as one of the research subjects, and showed some of the construction safety status of construction workers in China, filling the gaps in the international literature on construction safety in China. Second, this paper provides the first research on the combination of safe citizenship behavior and social motivation theory. After considering the four dimensions of safety citizenship behavior, different influence mechanisms of social motivation (social safety capital) were discovered and individual motivation (autonomous safety motivation). Third, a motivation-based mediating model was developed and validated. This model facilitates an understanding of psychosocial drivers that explain “how” and “why” such behaviors occur.

In addition to the theoretical implications, the findings in this research also have several practical implications. They can provide some useful insight into how to promote different types of safety citizenship behaviors within the construction industry. First, workers with higher social safety capital were verified to be more likely to engage in safety citizenship behavior. Therefore, managers should understand that improving or enhancing employees’ safety citizenship behaviors can increase the social safety capital of employees by enhancing communication between employees and fostering mutual trust among employees. Second, management should pay more attention to workers’ education levels and work experience to ensure a high level of safety citizenship behavior. Specifically, low-education workers need to be informed about the importance of safety citizenship, and workers with insufficient work experience can be trained in various ways to ensure that they can better understand the need for safe production.

### 5.3. Research Limitations and Future Directions

Although these findings are encouraging, this study still has some limitations. First, the questionnaire used the Wen Juan Xing app to collect data. While it has the advantage of having a widespread range and no physical position limitation, it cannot accurately determine whether the collected data come from the target population. Thus, the risk of invisible invalid questionnaires increases, which may affect the results of the data analysis. At the same time, the data come from the self-assessment of the respondent, and the results may differ from the actual situation.

Future SCB researchers should pay noticeable attention to collect objective data by collecting self-assessments and other assessments (such as co-workers and leaders); we can comprehensively collect the specific implementation of individuals’ safety and security behaviors during project construction, so that SCB can be measured more objectively. Second, the data obtained from the questionnaire survey were cross-sectional. However, the safety motivations affecting SCB and the formation and development of social safety capital are a continuous and clear sequence of processes. For the target population, the safety motivations and social safety capital displayed were in a static point data, not from the whole process of the data, which lost part of the interpretation of SCB. Future research can improve its data collection. Thus, although the mediating model at the core of the study stems from a sound theoretical basis, the results should be interpreted with caution. However, the evidence found will provide useful insights for supporting the possibility of further studies to confirm the mediating model longitudinally or through a diary study. Future studies could, for example, focus on social safety motivation with a preliminary survey and then autonomous safety motivation, safety citizenship behavior with repeated daily measures fitting the postulated causal ordering, thus providing more strong evidence on cause–effect relationships between the variables.

## Figures and Tables

**Figure 1 ijerph-17-00866-f001:**
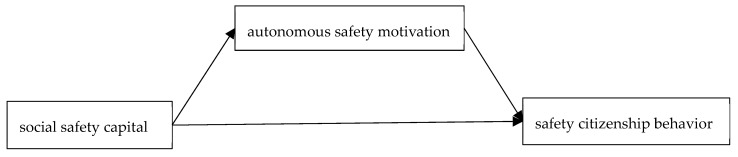
Framework of the mediated model proposed and tested in this study.

**Figure 2 ijerph-17-00866-f002:**
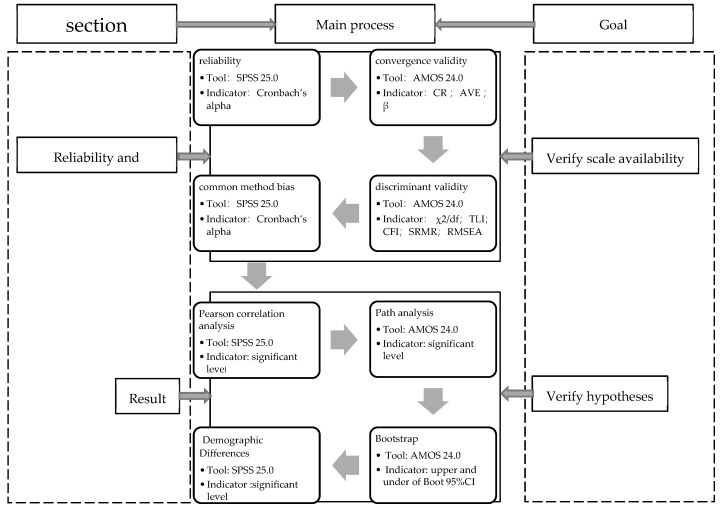
Data analysis flowchart.

**Figure 3 ijerph-17-00866-f003:**
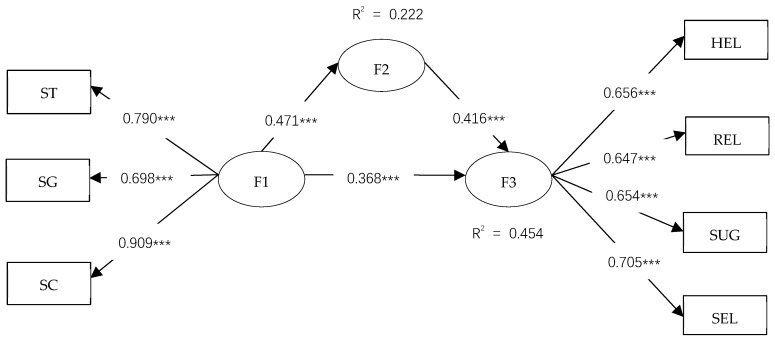
Results of the mediated model. Values on the paths are the standardized path coefficients. R^2^ represents the amount of variance the factor is accounted for in the model. *** At the 0.001 level, the correlation is significant. F1 is social safety capital, F2 is autonomous safety motivation, and F3 is safety citizenship behavior. HEL is mutual aid among workers; REL is the relationship between superior and subordinate; SUG is the participation of suggestion-making; SEL is self-control; SM is autonomous safety motivation; ST is safety trust; SG is safety goal; and SC is safety communication.

**Figure 4 ijerph-17-00866-f004:**
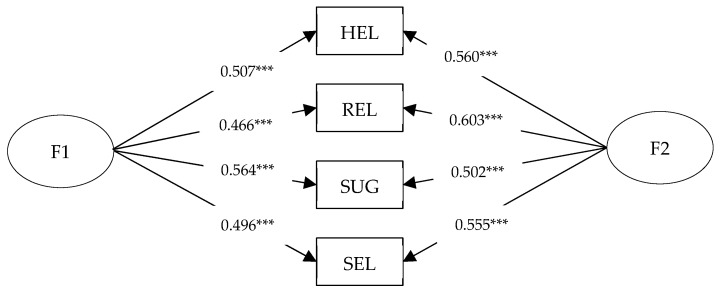
Different effects of social motivation/individual motivation. Among them, F1 is social safety capital and F2 is autonomous security motivation. *** At the 0.001 level, the correlation is significant. HEL is mutual aid among workers; REL is the relationship between superior and subordinate; SUG is participation of suggestion-making; and SEL is self-control.

**Table 1 ijerph-17-00866-t001:** Literature of the three dimensions of social capital.

Author	Relational	Cognitive	Structural	Nature of Research
Chow et al. [21]	social trust	shared goals	social network	knowledge-sharing
Zhou et al. [24]	——	social trust social reciprocity	social network social participation	antenatal depression
Chiu et al. [20,25]	trust, identification norm of reciprocity	shared vision shared language	social interaction ties	knowledge-sharing
Sauk et al. [26]	social trust	social goals	social tie	knowledge-sharing
Giordano et al. [27]	interpersonal trust	reciprocity	social participation	public health
Factors considered in our study	safety trust	safety goal	safety communication	construction safety

**Table 2 ijerph-17-00866-t002:** The sources of constructs.

Constructs		Abbreviations	Item Number	Reference
Safety citizenship behavior	Mutual aid among workers	HEL	3	[3,37]
	Relationship between superior and subordinate	REL	3	
	Participation of suggestion-making	SUG	3	
	Self-control	SEL	3	
Autonomous safety motivation	Autonomous safety motivation	SM	6	[29]
Social safety capital	Safety trust	ST	3	[20,26,27]
	Safety goal	SG	3	
	Safety communication	SC	4	

**Table 3 ijerph-17-00866-t003:** Multi-model confirmatory factor analysis (CFA) comparison test.

Model	χ2/df	TLI	CFI	RMSEA	SRMR	Model Comparison Test			
						Model Comparison	△χ2	Sig.	△df
Model 1	1.950	0.911	0.925	0.055	0.054				
Model 2	2.516	0.858	0.880	0.070	0.173	2 vs. 1	169.991	***	1
Model 3	2.513	0.858	0.880	0.070	0.178	3 vs. 1	169.149	***	1
Model 4	2.407	0.868	0.888	0.067	0.152	4 vs. 1	137.577	***	1
Model 5	2.268	0.881	0.899	0.064	0.133	5 vs. 1	96.360	***	1
Model 6	5.655	0.564	0.597	0.123	0.111	6 vs. 1	1254.971	***	28

Note: The factor in Model 1 is HEL, REL, SUG, SEL, ST, SG, SC, SM. The factor in Model 2 is HEL+REL, SUG, SEL, ST, SG, SC, SM. The factor in Model 3 is HEL, REL+SUG, SEL, ST, SG, SC, SM. The factor in Model 4 is HEL, REL, SUG, SEL+ST, SG, SC, SM. The factor in Model 5 is HEL, REL, SUG, SEL, ST, SG+SC, SM. Model 6 is a method factor that was only relevant to all items. HEL is mutual aid among workers; REL is the relationship between superior and subordinate; SUG is the participation of suggestion-making; SEL is self-control; SM is autonomous safety motivation; ST is safety trust; SG is safety goal; and SC is safety communication. Sig. is significant. *** At the 0.001 level, the output is significant. χ2/df is degrees of freedom, TLI is Tucker–Lewis index, CFI is comparative fit index, SRMR is standardized root mean squared residual, and RMSEA is root mean square error of approximation.

**Table 4 ijerph-17-00866-t004:** Validation factor analysis and reliability output results.

Constructs		Items	β	CR	AVE	Cronbach’s Alpha
safety citizenship behavior	HEL	HEL1	0.564	0.708	0.450	0.745
		HEL2	0.696			
		HEL3	0.740			
	REL	REL1	0.765	0.676	0.512	
		REL3	0.662			
	SUG	SUG1	0.639	0.784	0.552	
		SUG2	0.710			
		SUG3	0.862			
	SEL	SEL1	0.677	0.780	0.543	
		SEL2	0.699			
		SEL3	0.826			
social safety capital	ST	ST1	0.691	0.805	0.580	0.901
		ST2	0.791			
		ST3	0.797			
	SG	SG1	0.820	0.802	0.582	
		SG2	0.875			
		SG3	0.556			
	SC	SC1	0.748	0.864	0.614	
		SC2	0.799			
		SC3	0.782			
		SC4	0.803			
autonomous safety motivation		SM1	0.785	0.838	0.467	0.823
		SM2	0.524			
		SM3	0.631			
		SM4	0.743			
		SM5	0.703			
		SM6	0.681			

Note: CR is composite reliability. AVE is average variance extracted. β is normalized factor load. HEL is mutual aid among workers. REL is the relationship between superior and subordinate. SUG is the participation of suggestion-making. SEL is self-control. SM is autonomous safety motivation. ST is safety trust. SG is safety goal. SC is safety communication.

**Table 5 ijerph-17-00866-t005:** Descriptive statistics and correlation analysis of major dimensions.

	Mean	Standard Deviation	1. HEL	2. REL	3. SUG	4. SEL	5. ST	6. SG	7. SC	8. SM
1. HEL	4.543	0.571	1							
2. REL	4.614	0.577	0.395 **	1						
3. SUG	4.333	0.705	0.428 **	0.440 **	1					
4. SEL	4.684	0.504	0.515 **	0.474 **	0.400 **	1				
5. ST	4.239	0.614	0.227 **	0.205**	0.310 **	0.296 **	1			
6. SG	4.128	0.678	0.206 **	0.262 **	0.285 **	0.274 **	0.567 **	1		
7. SC	4.239	0.606	0.322 **	0.288 **	0.466 **	0.357 **	0.718 **	0.628 **	1	
8. SM	4.768	0.344	0.353 **	0.390 **	0.386 **	0.371 **	0.357 **	0.335 **	0.398 **	1

Note: ** At the 0.01 level (double-tailed), the correlation is significant. HEL is mutual aid among workers; REL is the relationship between superior and subordinate; SUG is the participation of suggestion-making; SEL is self-control; SM is autonomous safety motivation; ST is safety trust; SG is safety goal; and SC is safety communication.

**Table 6 ijerph-17-00866-t006:** The standardized direct, indirect, and total effects of social safety capital on safety citizenship behavior (SCB).

Effect Types	Effect Value	Boot SE	Z	Sig.	Boot 95% CI		Relative Effect
					Under	Upper	
Total effect	0.5645	0.0918	6.1495	***	0.3538	0.7115	
Direct effect	0.3684	0.1055	3.4915	***	0.1322	0.5489	65.3%
Indirect effect	0.1962	0.0448	4.3787	***	0.1190	0.3035	34.7%

Note: Sig. is significant. *** At the 0.001 level, the output is significant. Boot SE is standard error of mean by Bootstrap. under is the lowest value of the 95% confidence interval after the median effect test using the bootstrap method. upper is the highest value of the 95% confidence interval after the median effect test using the bootstrap method.

**Table 7 ijerph-17-00866-t007:** Results of ANOVA in terms of demographic information.

	Dependent Variable	HEL	REL	SUG	SEL
Feature					
gender		0.000 *	0.941	0.260	0.443
age		0.007 *	0.025 *	0.037 *	0.420
education		0.001 *	0.988	0.030 *	0.002 *
Work experience		0.016 *	0.311	0.432	0.975

Note: 1. HEL is mutual aid among workers; REL is the relationship between superior and subordinate; SUG is participation of suggestion-making; and SEL is self-control. 2. The data in the table are the Sig. of in the output result after Analysis of Variance. 3. The test level α is set to 0.05, and the statistically significant difference between the groups has been marked with *.

**Table 8 ijerph-17-00866-t008:** Non-standardized results of subgroup analysis of work experience by SPSS 25.0.

Subgroups	Percent (%)	Direct Effect	Indirect Effect	Boot LLCI	Boot ULCI	Model Sig.
<2	24.8	0.158	0.106	0.010	0.246	*
2–5	25.1	0.285 ***	0.122	0.019	0.225	***
6–10	19.6	0.312 ***	0.126	0.015	0.237	***
>10	30.5	0.267 ***	0.128	0.048	0.241	***
All sample	100	0.249 ***	0.125	0.074	0.180	***

Note: Sig. is significant. *** At the 0.001 level, the output is significant. * At the 0.05 level, the output is significant. Boot LLCI is the lowest value of the 95% confidence interval after the median effect test using the bootstrap method. Boot ULCI is the highest value of the 95% confidence interval after the median effect test using the bootstrap method.

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
