# Peer review of "Influence of Social Safety Capital on Safety Citizenship Behavior: The Mediation of Autonomous Safety Motivation"

_ijerph, 2020, doi:10.3390/ijerph17030866_

Round 1

Reviewer 1 Report

This paper is focused on the Influence of Social Safety Capital on Safety Citizenship Behavior and uses a questionnaire survey of the application herein. The subject of the manuscript falls within the scope of the journal and the results of the paper are of sufficiently high impact and global relevance for publication in an international journal. The statistical analysis seems to be highly developed and adequate. The questionnaire and hypotheses are described in general form and the framework of the study is described. Extensive statistical analysis is performed but in some form, they need to provide a better description of the questionnaire survey for the improvement of the paper quality. The interpretations and conclusions are adequate; they are justified by the data and consistent with the objectives. The organization of the article admits improvement in its presentation structure to include some relevant information on the questions asked and a better description of the respondents.

Specific points to be addressed are:

 - “The research model was used to study SCB, in which the target population of the questionnaire survey was construction workers, who worked in high-risk work environments and whose individual safety was more vulnerable.” What do you mean by high risk environment? What exactly is a more vulnerable safety? And what is the geographical locus of the population (country/region/etc.);

- The paper needs a better description of the questions made and the way it was applied and not just the statistical analysis of the data to allow full evaluation or understanding of the findings;

Reviewer 2 Report

This is a high quality paper. However, a number of modifications could improve it. The authors should reduce the introduction section, by better emphasis on the objective of the paper. The surplus text could be moved to the literature review sections. The authors need to debate the impact of the different ranges of respondents' experiences. Currently 50% of the respondents have 0-5 years of experience, while 30% have more than 10 years of experience. How does variation affect the significance of the obtained responses to the survey.    The authors should develop and include a set of recommendations pertaining to the findings of the research. These recommendations could be made available in the conclusion section.  The authors need to review the paper fully to correct the grammatical and syntax errors.

Reviewer 3 Report

This article evaluates the influence of social safety capital on safety citizenship behavior. It uses a motivation theory to explore the driving factor of social citizenship behavior from the perspective of social motivation and individual motivation. The work is in the scope of the journal, but redaction and structure should be improved as indicated below. Especially, the methods and results should be clearer; the author is recommended to identify and practice sophisticated objectives for this work. It was impossible for me to identify the novelty or even the actual scientific contribution of the paper. The author must justify the following points:

In the abstract, the author needs to illustrate the methods used, and highlight a concise conclusion drawn from the output results.

The paper should be revised to highlight novelties. Please consider that this lack of novelty starts with the Abstract, Introduction, and Conclusion. The aim of this work should be clarified more clearly. What are the importance and scientific contribution of this paper?

In line 58, what do you mean by “researchers have chosen”? who chooses? Please be more specific. In scientific papers, researches analyze not choose.

What is the database that SPSS 25 and AMOS 24 are using? It is highly important to explain the database and the mathematical Equations that the applied software is using in order to justify the collected results. This should be explained in the Methods Section.

In line 107, there is two “second”, please check it!

Section 2 should be built-in in Section 3 as it illustrates a part of the materials and methods of this work.

At the beginning of the Methods Section, I would suggest designing a Figure that illustrates the materials and methods used in this work. This Figure will help the readers understanding the construction and correlation of the methods used more easily.

As the questionnaire is the main source of information for this work, I suggest adding it as a supplementary document where the reader could make access and prove the discussion.

In Table 3, what do you mean by “Mean” and “SD”? Further, what are the applied Equations herein to conclude this sequence? The same is applicable in Table 4, Table 5, Table 6, Table 7, and Table 8. Identifying the Equations is an important issue to be justified for journal publication.

In Subsection 3.2.1., the author needs to explain this subsection in more detail. What do you mean by 0.7? what are the lowest and highest score values and what do they mean?

In Table 4, the author needs to explain all variables in the Table (i.e. hel, rel, sug, sel, ST, SG, SC, p, β, Cronbach’s α).

In Table 5, there is a need to better explain the factor in Models presented herein. What each column means and how was estimated? 

In lines 263 to 265, this is not the right place to explain and justify these variables. The same issue is applicable in Lines 285 to 287.

What are 1, 2, 3, 4, 5, 6, 7, and 8 stand for in Table 6?

What do you mean by Boot SE, Z, and p presented in Table 7?

In the conclusion section, the author needs to highlight the novelty and aims of the study again, as well as summarizing the applied methods. At the end of this section, there is a need to present the limitations and recommendations for future works. Besides, the author has to present the effective scientific contribution of this work. This aspect must be enhanced in conclusions.

The authors are using (we) too much. Please consider that this is a scientific journal publication, where you need to avoid some phrases like (we, our, ….). Instead, you can use (this work, this study, this analysis….). Besides, a proof reading by a native English speaker should be conducted to improve the clarity of the manuscript.

Round 2

Reviewer 1 Report

Paper is adequate.

Reviewer 2 Report

The revised paper has significantly been improved. All my corrections have been answered.

Reviewer 3 Report

The work has really developed and the author justified all my comments. I have no further comments to give and highly recommend this work for publicaiton.